# SerialRank: Spectral Ranking using Seriation

**Fajwel Fogel**
C.M.A.P., École Polytechnique,
Palaiseau, France
fogel@cmap.polytechnique.fr

**Alexandre d'Aspremont**
CNRS & D.I., École Normale Supérieure
Paris, France
aspremon@ens.fr

**Milan Vojnovic**
Microsoft Research,
Cambridge, UK
milanv@microsoft.com

## Abstract

We describe a seriation algorithm for ranking a set of $n$ items given pairwise comparisons between these items. Intuitively, the algorithm assigns similar rankings to items that compare similarly with all others. It does so by constructing a similarity matrix from pairwise comparisons, using seriation methods to reorder this matrix and construct a ranking. We first show that this spectral seriation algorithm recovers the true ranking when all pairwise comparisons are observed and consistent with a total order. We then show that ranking reconstruction is still exact even when some pairwise comparisons are corrupted or missing, and that seriation based spectral ranking is more robust to noise than other scoring methods. An additional benefit of the seriation formulation is that it allows us to solve semi-supervised ranking problems. Experiments on both synthetic and real datasets demonstrate that seriation based spectral ranking achieves competitive and in some cases superior performance compared to classical ranking methods.

## 1   Introduction

We study the problem of ranking a set of $n$ items given pairwise comparisons between these items. In practice, the information about pairwise comparisons is usually *incomplete*, especially in the case of a large set of items, and the data may also be *noisy*, that is some pairwise comparisons could be incorrectly measured and incompatible with the existence of a total ordering.

Ranking is a classic problem but its formulations vary widely. For example, website ranking methods such as PageRank [Page et al., 1998] and HITS [Kleinberg, 1999] seek to rank web pages based on the hyperlink structure of the web, where links do not necessarily express consistent preference relationships (e.g. $a$ can link to $b$ and $b$ can link $c$, and $c$ can link to $a$). The setting we study here goes back at least to [Kendall and Smith, 1940] and seeks to reconstruct a ranking between items from pairwise comparisons reflecting a total ordering.

In this case, the directed graph of all pairwise comparisons, where every pair of vertices is connected by exactly one of two possible directed edges, is usually called a *tournament* graph in the theoretical computer science literature or a "round robin" in sports, where every player plays every other player once and each preference marks victory or defeat. The motivation for this formulation often stems from the fact that in many applications, e.g. music, images, and movies, preferences are easier to express in relative terms (e.g. $a$ is better than $b$) rather than absolute ones (e.g. $a$ should be ranked fourth, and $b$ seventh).

Assumptions about how the pairwise preference information is obtained also vary widely. A subset of preferences is measured adaptively in [Ailon, 2011; Jamieson and Nowak, 2011], while [Negahban et al., 2012], for example, assume that preferences are observed iteratively, and [Freund et al., 2003] extract them at random. In other settings, the full preference matrix is observed, but is perturbed by noise: in e.g. [Bradley and Terry, 1952; Luce, 1959; Herbrich et al., 2006], a parametric model is assumed over the set of permutations, which reformulates ranking as a maximum likelihood problem.

Loss function and algorithmic approaches vary as well. Kenyon-Mathieu and Schudy [2007], for example, derive a PTAS for the minimum feedback arc set problem on tournaments, i.e. the problem of finding a ranking that minimizes the number of upsets (a pair of players where the player ranked lower on the ranking beats the player ranked higher). In practice, the complexity of this method is relatively high, and other authors [see e.g. Keener, 1993; Negahban et al., 2012] have been using spectral methods to produce more efficient algorithms (each pairwise comparison is understood as a link pointing to the preferred item). Simple scoring methods such as the point difference rule [Huber, 1963; Wauthier et al., 2013] produce efficient estimates at very low computational cost. Ranking has also been approached as a prediction problem, i.e. learning to rank [Schapire and Singer, 1998], with [Joachims, 2002] for example using support vector machines to learn a score function. Finally, in the Bradley-Terry-Luce framework, the maximum likelihood problem is usually solved using fixed point algorithms or EM-like majorization-minimization techniques [Hunter, 2004] for which no precise computational complexity bounds are known.

Here, we show that the ranking problem is directly related to another classical ordering problem, namely *seriation*: we are given a similarity matrix between a set of $n$ items and assume that the items can be ordered along a chain such that the similarity between items decreases with their distance within this chain (i.e. a total order exists). The seriation problem then seeks to reconstruct the underlying linear ordering based on unsorted, possibly noisy, pairwise similarity information. Atkins et al. [1998] produced a spectral algorithm that exactly solves the seriation problem in the noiseless case, by showing that for similarity matrices computed from serial variables, the ordering of the second eigenvector of the Laplacian matrix (a.k.a. the Fiedler vector) matches that of the variables. In practice, this means that spectral clustering exactly reconstructs the correct ordering provided items are organized in a chain. Here, adapting these results to ranking produces a very efficient *polynomial-time ranking algorithm with provable recovery and robustness guarantees*. Furthermore, the seriation formulation allows us to handle semi-supervised ranking problems. Fogel et al. [2013] show that seriation is equivalent to the 2-SUM problem and study convex relaxations to seriation in a semi-supervised setting, where additional structural constraints are imposed on the solution. Several authors [Blum et al., 2000; Feige and Lee, 2007] have also focused on the directly related Minimum Linear Arrangement (MLA) problem, for which excellent approximation guarantees exist in the noisy case, albeit with very high polynomial complexity.

The main contributions of this paper can be summarized as follows. We link seriation and ranking by showing how to construct a consistent similarity matrix based on consistent pairwise comparisons. We then recover the true ranking by applying the spectral seriation algorithm in [Atkins et al., 1998] to this similarity matrix (we call this method SerialRank in what follows). In the noisy case, we then show that spectral seriation can perfectly recover the true ranking even when some of the pairwise comparisons are either corrupted or missing, provided that the pattern of errors is relatively unstructured. We show in particular that, in a regime where a high proportion of comparisons are observed, some incorrectly, the spectral solution is more robust to noise than classical scoring based methods. Finally, we use the seriation results in [Fogel et al., 2013] to produce semi-supervised ranking solutions.

The paper is organized as follows. In Section 2 we recall definitions related to seriation, and link ranking and seriation by showing how to construct well ordered similarity matrices from well ranked items. In Section 3 we apply the spectral algorithm of [Atkins et al., 1998] to reorder these similarity matrices and reconstruct the true ranking in the noiseless case. In Section 4 we then show that this spectral solution remains exact in a noisy regime where a random subset of comparisons is corrupted. Finally, in Section 5 we illustrate our results on both synthetic and real datasets, and compare ranking performance with classical maximum likelihood, spectral and scoring based approaches. Auxiliary technical results are detailed in Appendix A.

## 2 Seriation, Similarities & Ranking

In this section we first introduce the seriation problem, i.e. reordering items based on pairwise similarities. We then show how to write the problem of ranking given pairwise comparisons as a seriation problem.

### 2.1 The Seriation Problem

The seriation problem seeks to reorder $n$ items given a similarity matrix between these items, such that the more similar two items are, the closer they should be. This is equivalent to supposing that items can be placed on a chain where the similarity between two items decreases with the distance between these items in the chain. We formalize this below, following [Atkins et al., 1998].

**Definition 2.1** *We say that the matrix $A \in \mathbf{S}_n$ is an R-matrix (or Robinson matrix) if and only if it is symmetric and $A_{i,j} \leq A_{i,j+1}$ and $A_{i+1,j} \leq A_{i,j}$ in the lower triangle, where $1 \leq j < i \leq n$.*

Another way to formulate R-matrix conditions is to impose $A_{ij} \leq A_{kl}$ if $|i - j| \leq |k - l|$ off-diagonal, i.e. the coefficients of $A$ decrease as we move away from the diagonal. We also introduce a definition for strict R-matrices $A$, whose rows/columns cannot be permuted without breaking the R-matrix monotonicity conditions. We call *reverse identity* permutation the permutation that puts rows and columns $\{1, \ldots, n\}$ of a matrix $A$ in reverse order $\{n, n-1, \ldots, 1\}$.

**Definition 2.2** *An R-matrix $A \in \mathbf{S}_n$ is called strict-R if and only if the identity and reverse identity permutations of $A$ are the only permutations producing R-matrices.*

Any R-matrix with only strict R-constraints is a strict R-matrix. Following [Atkins et al., 1998], we will say that $A$ is *pre-R* if there is a permutation matrix $\Pi$ such that $\Pi A \Pi^T$ is a R-matrix. Given a pre-R matrix $A$, the seriation problem consists in finding a permutation $\Pi$ such that $\Pi A \Pi^T$ is a R-matrix. Note that there might be several solutions to this problem. In particular, if a permutation $\Pi$ is a solution, then the reverse permutation is also a solution. When only two permutations of $A$ produce R-matrices, $A$ will be called *pre-strict-R*.

### 2.2 Constructing Similarity Matrices from Pairwise Comparisons

Given an ordered input pairwise comparison matrix, we now show how to construct a similarity matrix which is *strict-R* when all comparisons are given and consistent with the identity ranking (*i.e.* items are ranked in the increasing order of indices). This means that the similarity between two items decreases with the distance between their ranks. We will then be able to use the spectral seriation algorithm by [Atkins et al., 1998] described in Section 3 to recover the true ranking from a disordered similarity matrix.

We first explain how to compute a pairwise similarity from binary comparisons between items by counting the number of matching comparisons. Another formulation allows to handle the generalized linear model.

#### 2.2.1 Similarities from Pairwise Comparisons

Suppose we are given a matrix of pairwise comparisons $C \in \{-1, 0, 1\}^{n \times n}$ such that $C_{i,j} + C_{j,i} = 0$ for every $i \neq j$ and

$$C_{i,j} = \begin{cases} 1 & \text{if } i \text{ is ranked higher than } j \\ 0 & \text{if } i \text{ and } j \text{ are not compared or in a draw} \\ -1 & \text{if } j \text{ is ranked higher than } i \end{cases} \tag{1}$$

and, by convention, we define $C_{i,i} = 1$ for all $i \in \{1, \ldots, n\}$ ($C_{i,i}$ values have no effect in the ranking method presented in algorithm SerialRank). We also define the pairwise similarity matrix $S^{\mathrm{match}}$ as

$$S_{i,j}^{\mathrm{match}} = \sum_{k=1}^n \left( \frac{1 + C_{i,k} C_{j,k}}{2} \right). \tag{2}$$

Since $C_{i,k}C_{j,k} = 1$ if $C_{i,k}$ and $C_{j,k}$ have same signs, and $C_{i,k}C_{j,k} = -1$ if they have opposite signs, $S_{i,j}^{\text{match}}$ counts the number of matching comparisons between $i$ and $j$ with other reference items $k$. If $i$ or $j$ is not compared with $k$, then $C_{i,k}C_{j,k} = 0$ and the term $(1 + C_{i,k}C_{j,k})/2$ has an average effect on the similarity of $1/2$. The intuition behind this construction is easy to understand in a tournament setting: players that beat the same players and are beaten by the same players should have a similar ranking. We can write $S^{\text{match}}$ in the following equivalent form

$$S^{\text{match}} = \frac{1}{2}\left(n\mathbf{1}\mathbf{1}^T + CC^T\right). \tag{3}$$

Without loss of generality, we assume in the following propositions that items are ranked in increasing order of their indices (identity ranking). In the general case, we simply replace the *strict-R* property by the *pre-strict-R* property.

The next result shows that when all comparisons are given and consistent with the identity ranking, then the similarity matrix $S^{\text{match}}$ is a strict R-matrix.

**Proposition 2.3** *Given all pairwise comparisons $C_{i,j} \in \{-1, 0, 1\}$ between items ranked according to the identity permutation (with no ties), the similarity matrix $S^{\text{match}}$ constructed as given in (2) is a strict R-matrix and*

$$S_{ij}^{\text{match}} = n - (\max\{i, j\} - \min\{i, j\}) \tag{4}$$

*for all $i, j = 1, \ldots, n$.*

### 2.2.2 Similarities in the Generalized Linear Model

Suppose that paired comparisons are generated according to a generalized linear model (GLM), *i.e.* we assume that the outcomes of paired comparisons are independent and for any pair of distinct items, item $i$ is observed to be preferred over item $j$ with probability

$$P_{i,j} = H(\nu_i - \nu_j) \tag{5}$$

where $\nu \in \mathbb{R}^n$ is a vector of strengths or skills parameters and $H : \mathbb{R} \to [0, 1]$ is a function that is increasing on $\mathbb{R}$ and such that $H(-x) = 1 - H(x)$ for all $x \in \mathbb{R}$, and $\lim_{x\to-\infty} H(x) = 0$ and $\lim_{x\to\infty} H(x) = 1$. A well known special instance of the generalized linear model is the Bradley-Terry-Luce model for which $H(x) = 1/(1 + e^{-x})$, for $x \in \mathbb{R}$.

Let $m_{i,j}$ be the number of times items $i$ and $j$ were compared, $C_{i,j}^s \in \{-1, 1\}$ be the outcome of comparison $s$ and $Q$ be the matrix of corresponding empirical probabilities, i.e. if $m_{i,j} > 0$ we have

$$Q_{i,j} = \frac{1}{m_{i,j}} \sum_{s=1}^{m_{i,j}} \frac{C_{i,j}^s + 1}{2}$$

and $Q_{i,j} = 1/2$ in case $m_{i,j} = 0$. We then define the similarity matrix $S^{\text{glm}}$ from the observations $Q$ as

$$S_{i,j}^{\text{glm}} = \sum_{k=1}^{n} \mathbb{1}_{\{m_{i,k}m_{j,k}>0\}}\left(1 - \frac{|Q_{i,k} - Q_{j,k}|}{2}\right) + \frac{\mathbb{1}_{\{m_{i,k}m_{j,k}=0\}}}{2}. \tag{6}$$

Since the comparisons are independent we have that $Q_{i,j}$ converges to $P_{i,j}$ as $m_{i,j}$ goes to infinity and

$$S_{i,j}^{\text{glm}} \to \sum_{k=1}^{n}\left(1 - \frac{|P_{i,k} - P_{j,k}|}{2}\right).$$

The result below shows that this limit similarity matrix is a strict R-matrix when the variables are properly ordered.

**Proposition 2.4** *If the items are ordered according to the order in decreasing values of the skill parameters, in the limit of large number of observations, the similarity matrix $S^{\text{glm}}$ is a strict R matrix.*

Notice that we recover the original definition of $S^{\text{match}}$ in the case of binary probabilities, though it does not fit in the Generalized Linear Model. Note also that these definitions can be directly extended to the setting where multiple comparisons are available for each pair and aggregated in comparisons that take fractional values (*e.g.* in a tournament setting where participants play several times against each other).

---

**Algorithm 1** Using Seriation for Spectral Ranking (SerialRank)

---

**Input:** A set of pairwise comparisons $C_{i,j} \in \{-1, 0, 1\}$ or $[-1, 1]$.

  1: Compute a similarity matrix $S$ as in §2.2
  2: Compute the Laplacian matrix

$$L_S = \mathbf{diag}(S\mathbf{1}) - S \qquad\qquad \text{(SerialRank)}$$

  3: Compute the Fiedler vector of $S$.
**Output:** A ranking induced by sorting the Fiedler vector of $S$ (choose either increasing or decreasing order to minimize the number of upsets).

---

## 3 Spectral Algorithms

We first recall how the spectral clustering approach can be used to recover the true ordering in seriation problems by computing an eigenvector, with computational complexity $O(n^2 \log n)$ [Kuczynski and Wozniakowski, 1992]. We then apply this method to the ranking problem.

### 3.1 Spectral Seriation Algorithm

We use the spectral computation method originally introduced in [Atkins et al., 1998] to solve the seriation problem based on the similarity matrices defined in the previous section. We first recall the definition of the Fiedler vector.

**Definition 3.1** *The Fiedler value of a symmetric, nonnegative and irreducible matrix $A$ is the smallest non-zero eigenvalue of its Laplacian matrix $L_A = \mathbf{diag}(A\mathbf{1}) - A$. The corresponding eigenvector is called Fiedler vector and is the optimal solution to $\min\{y^T L_A y : y \in \mathbb{R}^n, y^T \mathbf{1} = 0, \|y\|_2 = 1\}$.*

The main result from [Atkins et al., 1998], detailed below, shows how to reorder pre-R matrices in a noise free case.

**Proposition 3.2** *[Atkins et al., 1998, Th. 3.3] Let $A \in \mathbf{S}_n$ be an irreducible pre-R-matrix with a simple Fiedler value and a Fiedler vector $v$ with no repeated values. Let $\Pi_1 \in \mathcal{P}$ (respectively, $\Pi_2$) be the permutation such that the permuted Fiedler vector $\Pi_1 v$ is strictly increasing (decreasing). Then $\Pi_1 A \Pi_1^T$ and $\Pi_2 A \Pi_2^T$ are R-matrices, and no other permutations of $A$ produce R-matrices.*

### 3.2 SerialRank: a Spectral Ranking Algorithm

In Section 2, we showed that similarities $S^{\mathrm{match}}$ and $S^{\mathrm{glm}}$ are *pre-strict-R* when all comparisons are available and consistent with an underlying ranking of items. We now use the spectral seriation method in [Atkins et al., 1998] to reorder these matrices and produce an output ranking. We call this algorithm SerialRank and prove the following result.

**Proposition 3.3** *Given all pairwise comparisons for a set of totally ordered items and assuming there are no ties between items, performing algorithm SerialRank, i.e. sorting the Fiedler vector of the matrix $S^{\mathrm{match}}$ defined in (3) recovers the true ranking of items.*

Similar results apply for $S^{\mathrm{glm}}$ when we are given enough comparisons in the Generalized Linear Model. This last result guarantees recovery of the true ranking of items in the noiseless case. In the next section, we will study the impact of corrupted or missing comparisons on the inferred ranking of items.

### 3.3 Hierarchical Ranking

In a large dataset, the goal may be to rank only a subset of top rank items. In this case, we can first perform spectral ranking (cheap) and then refine the ranking of the top set of items using either the SerialRank algorithm on the top comparison submatrix, or another seriation algorithm such as

the convex relaxation in [Fogel et al., 2013]. This last method would also allow us to solve semi-supervised ranking problems, given additional information on the structure of the solution.

## 4   Robustness to Corrupted and Missing Comparisons

In this section we study the robustness of SerialRank using $S^{\mathrm{match}}$ with respect to noisy and missing pairwise comparisons. We will see that noisy comparisons cause ranking ambiguities for the standard point score method and that such ambiguities can be lifted by the spectral ranking algorithm. We show in particular that the SerialRank algorithm recovers the exact ranking when the pattern of errors is random and errors are not too numerous.

We define here the *point score $w_i$* of an item $i$, also known as point-difference, or *row-sum*, as $w_i = \sum_{k=1}^{n} C_{k,i}$ which corresponds to the number of wins minus the number of losses in a tournament setting.

**Proposition 4.1** *Given all pairwise comparisons $C_{s,t} \in \{-1, 1\}$ between items ranked according to their indices, suppose the signs of $m$ comparisons indexed $(i_1, j_1), \ldots, (i_m, j_m)$ are switched.*

1. *For the case of one corrupted comparison, if $j_1 - i_1 > 2$ then the spectral ranking recovers the true ranking whereas the standard point score method induces ties between the pairs of items $(i_1, i_1 + 1)$ and $(j_1 - 1, j_1)$.*

2. *For the general case of $m \geq 1$ corrupted comparisons, suppose that the following condition holds true*

$$|i - j| > 2, \text{ for all } i, j \in \{i_1, \ldots, i_m, j_1, \ldots, j_m\} \text{ such that } i \neq j, \qquad (7)$$

*then, $S^{\mathrm{match}}$ is a strict R-matrix, and thus the spectral ranking recovers the true ranking whereas the standard point score method induces ties between $2m$ pairs of items.*

For the case of one corrupted comparison, note that the separation condition on the pair of items $(i, j)$ is necessary. When the comparison $C_{i,j}$ between two adjacent items according to the true ranking is corrupted, no ranking method can break the resulting tie. For the case of arbitrary number of corrupted comparisons, condition (7) is a sufficient condition only.

Using similar arguments, we can also study conditions for recovering the true ranking in the case with missing comparisons. These scenarios are actually slightly less restrictive than the noisy cases and are covered in the supplementary material. We now estimate the number of randomly corrupted entries that can be tolerated for perfect recovery of the true ranking.

**Proposition 4.2** *Given a comparison matrix for a set of $n$ items with $m$ corrupted comparisons selected uniformly at random from the set of all possible item pairs. Algorithm SerialRank guarantees that the probability of recovery $p(n, m)$ satisfies $p(n, m) \geq 1 - \delta$, provided that $m = O(\sqrt{\delta n})$. In particular, this implies that $p(n, m) = 1 - o(1)$ provided that $m = o(\sqrt{n})$.*

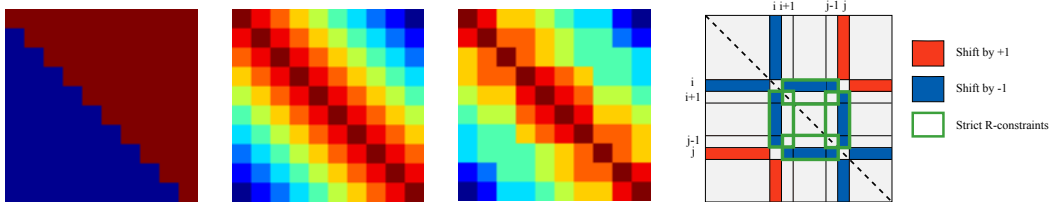

Figure 1: The matrix of pairwise comparisons $C$ *(far left)* when the rows are ordered according to the true ranking. The corresponding similarity matrix $S^{\mathrm{match}}$ is a strict R-matrix *(center left)*. The same $S^{\mathrm{match}}$ similarity matrix with comparison (3,8) corrupted *(center right)*. With one corrupted comparison, $S^{\mathrm{match}}$ keeps enough strict R-constraints to recover the right permutation. In the noiseless case, the difference between all coefficients is at least one and after introducing an error, the coefficients inside the green rectangles still enforce strict R-constraints *(far right)*.

# 5   Numerical Experiments

We conducted numerical experiments using both synthetic and real datasets to compare the performance of SerialRank with several classical ranking methods.

**Synthetic Datasets**   The first synthetic dataset consists of a binary matrix of pairwise comparisons derived from a given ranking of $n$ items with uniform, randomly distributed corrupted or missing entries. A second synthetic dataset consists of a full matrix of pairwise comparisons derived from a given ranking of $n$ items, with added uncertainty for items which are sufficiently close in the true ranking of items. Specifically, given a positive integer $m$, we let $C_{i,j} = 1$ if $i < j - m$, $C_{i,j} \sim \text{Unif}[-1, 1]$ if $|i-j| \leq m$, and $C_{i,j} = -1$ if $i > j+m$. In Figure 2, we measure the Kendall $\tau$ correlation coefficient between the true ranking and the retrieved ranking, when varying either the percentage of corrupted comparisons or the percentage of missing comparisons. Kendall's $\tau$ counts the number of agreeing pairs minus the number of disagreeing pairs between two rankings, scaled by the total number of pairs, so that it takes values between -1 and 1. Experiments were performed with $n = 100$ and reported Kendall $\tau$ values were averaged over 50 experiments, with standard deviation less than 0.02 for points of interest (*i.e.* here with Kendall $\tau > 0.8$).

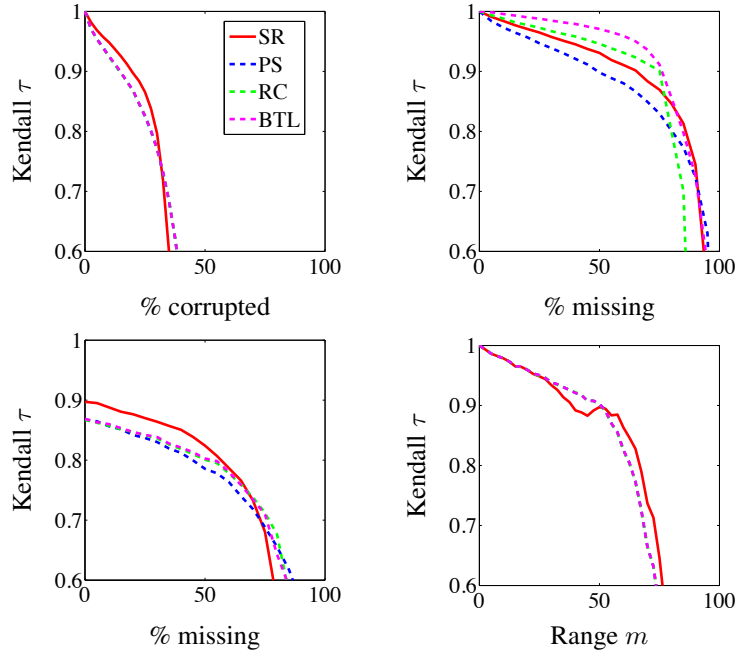

Figure 2: Kendall $\tau$ (higher is better) for SerialRank (SR, full red line), row-sum (PS, [Wauthier et al., 2013] dashed blue line), rank centrality (RC [Negahban et al., 2012] dashed green line), and maximum likelihood (BTL [Bradley and Terry, 1952], dashed magenta line). In the first synthetic dataset, we vary the proportion of corrupted comparisons *(top left)*, the proportion of observed comparisons *(top right)* and the proportion of observed comparisons, with 20% of comparisons being corrupted *(bottom left)*. We also vary the parameter $m$ in the second synthetic dataset *(bottom right)*.

**Real Datasets**   The first real dataset consists of pairwise comparisons derived from outcomes in the TopCoder algorithm competitions. We collected data from 103 competitions among 2742 coders over a period of about one year. Pairwise comparisons are extracted from the ranking of each competition and then averaged for each pair. TopCoder maintains ratings for each participant, updated in an online scheme after each competition, which were also included in the benchmarks. To measure performance in Figure 3, we compute the percentage of upsets (i.e. comparisons disagreeing with the computed ranking), which is closely related to the Kendall $\tau$ (by an affine transformation if comparisons were coming from a consistent ranking). We refine this metric by considering only the participants appearing in the top $k$, for various values of $k$, i.e. computing

$$l_k = \frac{1}{|\mathcal{C}_k|} \sum_{i,j \in \mathcal{C}_k} \mathbb{1}_{\{r(i)>r(j)\}} \mathbb{1}_{\{C_{i,j}<0\}}, \tag{8}$$

where $\mathcal{C}$ are the pairs $(i,j)$ that are compared and such that $i,j$ are both ranked in the top $k$, and $r(i)$ is the rank of $i$. Up to scaling, this is the loss considered in [Kenyon-Mathieu and Schudy, 2007].

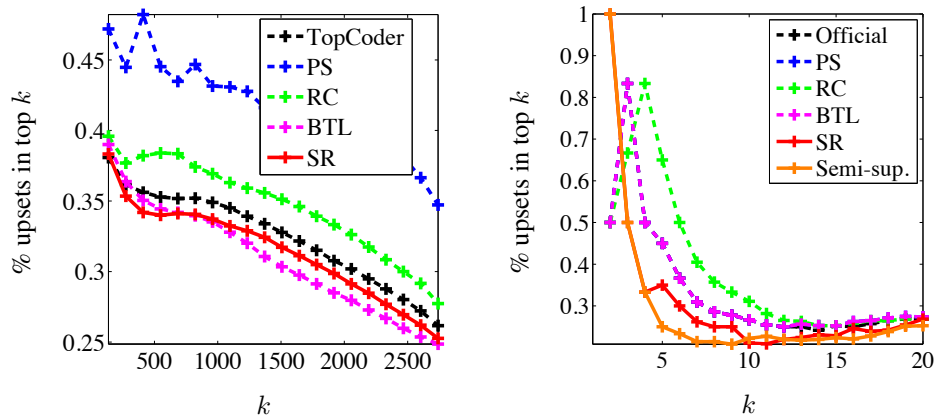

Figure 3: Percentage of upsets (i.e. disagreeing comparisons, lower is better) defined in (8), for various values of $k$ and ranking methods, on TopCoder (*left*) and football data (*right*).

**Semi-Supervised Ranking** We illustrate here how, in a semi-supervised setting, one can interactively enforce some constraints on the retrieved ranking, using e.g. the semi-supervised seriation algorithm in [Fogel et al., 2013]. We compute rankings of England Football Premier League teams for season 2013-2014 (*cf.* figure 4 in Appendix for previous seasons). Comparisons are defined as the averaged outcome (win, loss, or tie) of home and away games for each pair of teams. As shown in Table 1, the top half of SerialRank ranking is very close to the official ranking calculated by sorting the sum of points for each team (3 points for a win, 1 point for a tie). However, there are significant variations in the bottom half, though the number of upsets is roughly the same as for the official ranking. To test semi-supervised ranking, suppose for example that we are not satisfied with the ranking of Aston Villa (last team when ranked by the spectral algorithm), we can explicitly enforce that Aston Villa appears before Cardiff, as in the official ranking. In the ranking based on the semi-supervised corresponding seriation problem, Aston Villa is not last anymore, though the number of disagreeing comparisons remains just as low (*cf.* Figure 3, *right*).

Table 1: Ranking of teams in the England premier league season 2013-2014.

| Official | Row-sum | RC | BTL | SerialRank | Semi-Supervised |
|---|---|---|---|---|---|
| Man City (86) | Man City | Liverpool | Man City | Man City | Man City |
| Liverpool (84) | Liverpool | Arsenal | Liverpool | Chelsea | Chelsea |
| Chelsea (82) | Chelsea | Man City | Chelsea | Liverpool | Liverpool |
| Arsenal (79) | Arsenal | Chelsea | Arsenal | Arsenal | Everton |
| Everton (72) | Everton | Everton | Everton | Everton | Arsenal |
| Tottenham (69) | Tottenham | Tottenham | Tottenham | Tottenham | Tottenham |
| Man United (64) | Man United | Man United | Man United | Southampton | Man United |
| Southampton (56) | Southampton | Southampton | Southampton | Man United | Southampton |
| Stoke (50) | Stoke | Stoke | Stoke | Stoke | Newcastle |
| Newcastle (49) | Newcastle | Newcastle | Newcastle | Swansea | Stoke |
| Crystal Palace (45) | Crystal Palace | Swansea | Crystal Palace | Newcastle | West Brom |
| Swansea (42) | Swansea | Crystal Palace | Swansea | West Brom | Swansea |
| West Ham (40) | West Brom | West Ham | West Brom | Hull | Crystal Palace |
| Aston Villa (38) | West Ham | Hull | West Ham | West Ham | Hull |
| Sunderland (38) | Aston Villa | Aston Villa | Aston Villa | Cardiff | West Ham |
| Hull (37) | Sunderland | West Brom | Sunderland | Crystal Palace | Fulham |
| West Brom (36) | Hull | Sunderland | Hull | Fulham | Norwich |
| Norwich (33) | Norwich | Fulham | Norwich | Norwich | Sunderland |
| Fulham (32) | Fulham | Norwich | Fulham | Sunderland | Aston Villa |
| Cardiff (30) | Cardiff | Cardiff | Cardiff | Aston Villa | Cardiff |

**Acknowledgments** FF, AA and MV would like to acknowledge support from a European Research Council starting grant (project SIPA) and support from the MSR-INRIA joint centre.

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
