[Supplementary Material]

# A  Supplementary Material

## A.1  Similarity Matrices

We begin here by detailing proofs for the main results of Section 2.

**Proposition A.1** *Given all pairwise comparisons $C_{i,j} \in \{-1, 0, 1\}$ between items ranked according to the identity permutation (with no ties), the similarity matrix $S^{\mathrm{match}}$ constructed as given in (2) is a strict R-matrix and*

$$S_{ij}^{\mathrm{match}} = n - (\max\{i, j\} - \min\{i, j\}) \tag{9}$$

*for all $i, j \in \{1, \dots, n\}$.*

**Proof.** Since items are ranked as $\{1, \dots, n\}$ with no ties and all comparisons given, $C_{i,j} = -1$ if $i < j$ and $C_{i,j} = 1$ otherwise. Therefore we get from definition (2)

$$
\begin{aligned}
S_{i,j}^{\mathrm{match}} &= \sum_{k=1}^{\min(i,j)-1} \left(\frac{1+1}{2}\right) + \sum_{k=\min(i,j)}^{\max(i,j)-1} \left(\frac{1-1}{2}\right) + \sum_{k=\max(i,j)}^{n} \left(\frac{1+1}{2}\right) \\
&= n - (\max\{i, j\} - \min\{i, j\}).
\end{aligned}
$$

This means in particular that $S^{\mathrm{match}}$ is strictly positive and its coefficients are strictly decreasing when moving away from the diagonal, hence $S^{\mathrm{match}}$ is a strict R-matrix. Formally, using equation (4), we have for any $i < j$

$$S_{i,j}^{\mathrm{match}} = n - (\max\{i, j\} - \min\{i, j\}) = n - j + i > n - (j+1) + i = S_{i,j+1}^{\mathrm{match}},$$

and similarly $S_{i+1,j}^{\mathrm{match}} > S_{i,j}^{\mathrm{match}}$, which proves that $S^{\mathrm{match}}$ is a strict R-matrix. ∎

**Proposition A.2** *When the variables are ordered according to the order given by $\nu$, and the number of observations is large enough, then $S^{\mathrm{glm}}$ is a strict R matrix.*

**Proof.** Without loss of generality, we suppose the true order is $\{1, \dots, n\}$, with $\nu(1) > \dots > \nu(n)$. For any $i, j, k$ such that $i > j$, using the GLM assumption (i) we get

$$P_{i,k} = H(\nu(i) - \nu(k)) > H(\nu(j) - \nu(k)) = p_{j,k}.$$

Since empirical probabilities $Q_{ij}$ converge to $p_{ij}$, when the number of observations is large enough, we also get $Q_{i,k} > Q_{j,k}$ for any $i, j, k$ such that $i > j \geq k$ (we focus wlog on the lower triangle), and we can therefore remove the absolute value in the expression of $S_{ij}^{\mathrm{glm}}$ for $i > j$. Hence for any $i > j$ we have

$$
\begin{aligned}
S_{i+1,j}^{\mathrm{glm}} - S_{i,j}^{\mathrm{glm}} &= \frac{1}{2}\left(-\sum_{k=1}^{n} |Q_{i+1,k} - Q_{j,k}| + \sum_{k=1}^{n} |Q_{i,k} - Q_{j,k}|\right) \\
&= \frac{1}{2}\left(\sum_{k=1}^{n} -(Q_{i+1,k} - Q_{j,k}) + (Q_{i,k} - Q_{j,k})\right) \\
&= \frac{1}{2}\left(\sum_{k=1}^{n} Q_{i,k} - Q_{i+1,k}\right) < 0.
\end{aligned}
$$

Similarly for any $i > j$, $S_{i,j-1}^{\mathrm{glm}} - S_{i,j}^{\mathrm{glm}} < 0$, so $S^{\mathrm{glm}}$ is a strict R-matrix. ∎

## A.2  Spectral Algorithm

The next technical lemmas extend the results in Atkins et al. [1998]. The first one shows that without loss of generality, the Fiedler value is simple.

**Lemma A.3** *If $A$ is an irreducible R-matrix, up to a uniform shift of its coefficients, $A$ has a simple Fiedler value and a monotonic Fiedler vector.*

**Proof.** We use [Atkins et al., 1998, Th. 4.6] which states that if $A$ is an irreducible R-matrix with $A_{n,1} = 0$, then the Fiedler value of $A$ is a simple eigenvalue. Since $A$ is a R-matrix, $A_{n,1}$ is among its minimal elements. Subtracting it from $A$ does not affect the positivity of $A$ and we can apply [Atkins et al., 1998, Th. 4.6]. Monotonicity of the Fiedler vector then follows from [Atkins et al., 1998, Th. 3.2]. ∎

The next lemma shows that the Fiedler vector is strictly monotonic if $A$ is a strict R-matrix.

**Lemma A.4** *Let $A \in \mathbf{S}_n$ be a R-matrix. Suppose there are no distinct indices $r < s$ such that for any $k \notin [r; s]$, $A_{r,k} = A_{r+1,k} = \ldots = A_{s,k}$, then, up to a uniform shift, the Fiedler value of $A$ is simple and its Fiedler vector is strictly monotonic.*

**Proof.** By Lemma A.3, the Fiedler value of $A$ is simple (up to a uniform shift of $A$). Let $x$ be the corresponding Fiedler vector of $A$, $x$ is monotonic by Lemma A.3. Suppose $[r; s]$ is a nontrivial maximal interval such that $x_r = x_{r+1} = \ldots = x_s$, then by [Atkins et al., 1998, lemma 4.3], for any $k \notin [r; s]$, $A_{r,k} = A_{r+1,k} = \ldots = A_{s,k}$, which contradicts the initial assumption. Therefore $x$ is strictly monotonic. ∎

We now show that the condition of A.4 on $A$ are equivalent to A being strict-R.

**Lemma A.5** *An R-matrix $A \in \mathbf{S}_n$ is strictly R if and only if there are no distinct indices $r < s$ such that for any $k \notin [r; s]$, $A_{r,k} = A_{r+1,k} = \ldots = A_{s,k}$.*

**Proof.** Let $A \in \mathbf{S}_n$ a R-matrix. Let us first suppose there are no distinct indices $r < s$ such that for any $k \notin [r; s]$, $A_{r,k} = A_{r+1,k} = \ldots = A_{s,k}$. By lemma A.4 the Fiedler value of $A$ is simple and its Fiedler vector is strictly monotonic. Hence by proposition 3.2, only the identity and reverse identity permutations of $A$ produce R-matrices. Now suppose there exist two distinct indices $r < s$ such that for any $k \notin [r; s]$, $A_{r,k} = A_{r+1,k} = \ldots = A_{s,k}$. In addition to the identity and reverse identity permutations, we can locally reverse the order of rows and columns from $r$ to $s$, since the sub matrix $A_{r:s,r:s}$ is an R-matrix and for any $k \notin [r; s]$, $A_{r,k} = A_{r+1,k} = \ldots = A_{s,k}$. Therefore at least four different permutations of $A$ produce R-matrices, which means that $A$ is not strictly R. ∎

Combining previous lemma we obtain the main result of this section.

**Proposition A.6** *Given all pairwise comparisons between totally ordered variables and assuming there are no ties between items, performing algorithm SerialRank, i.e. sorting the Fiedler vector of the matrix $S^{\mathrm{match}}$ defined in (3) recovers the true ranking.*

**Proof.** From Proposition 2.3 we get that, under our assumptions, $S^{\mathrm{match}}$ is a pre-strict R-matrix. Now combining the equivalent definition of strict-R matrices in lemma A.5 with lemma A.4, we deduce that Fiedler value of $S^{\mathrm{match}}$ is simple and its Fiedler vector has no repeated values. Hence by theorem 3.2, only the two permutations that sort the Fiedler vector in increasing and decreasing order produce strict R-matrices and are therefore candidate rankings (since from Proposition 2.3 $S^{\mathrm{match}}$ is a strictly R-matrix when ordered according to the true ranking). Finally we can choose between the two candidate rankings (increasing and decreasing) by picking the one with the least upstets. ∎

## A.3 Robustness to Corrupted and Missing Comparisons

We show that $S^{\mathrm{match}}$ remains *pre-strict-R* even when the comparison matrix originally derived from a total order has a random pattern of errors, whereas the point score vectors defined at the beginning of Section 4 has ties. Then using the same argument as in the proof of proposition A.6, we deduce that algorithm SerialRank, *i.e.* sorting the Fiedler vector of the matrix $S^{\mathrm{match}}$ defined in (3) recovers the exact ranking when the pattern of errors is random (proposition 4.1). Similar results are derived for missing comparisons.

**Proposition A.7** *Given all pairwise comparisons $C_{s,t} \in \{-1, 1\}$ between items ranked according to their indices, suppose the sign of one comparison $C_{i,j}$ is switched, with $i < j$. If $j - i > 2$ then $S^{\mathrm{match}}$ defined in* (3) *remains* strict-R, *whereas the score vector $w$ has ties between items $i$ and $i+1$ and items $j$ and $j - 1$.*

**Proof.** We give some intuition on the result in Figure 1. We write the true score and comparison matrix $w$ and $C$, while the observations are written $\hat{w}$ and $\hat{C}$ respectively. This means in particular that $\hat{C}_{i,j} = -C_{i,j} = 1$. To simplify notations we denote by $S$ the similarity matrix $S^{\mathrm{match}}$ (respectively $\hat{S}$ when the similarity is computed from observations). We first study the impact of a corrupted comparison $C_{i,j}$ for $i < j$ on the score vector $\hat{w}$. We have

$$\hat{w}_i = \sum_{k=1}^{n} \hat{C}_{k,i} = \sum_{k=1}^{n} C_{k,i} + \hat{C}_{j,i} - C_{j,i} = w_i + 2 = w_{i+1},$$

similarly $\hat{w}_j = w_{j-1}$, whereas for $k \neq i, j$, $\hat{w}_k = w_k$. Hence, the incorrect comparison induces two ties in the score vector $w$.

Now we show that the similarity matrix defined in (3) breaks these ties, by showing that it is a strict R-matrix. Writing $\hat{S}$ in terms of $S$, we get

$$[\hat{C}\hat{C}^T]_{i,t} = \sum_{k \neq j} \left( \hat{C}_{i,k}\hat{C}_{t,k} \right) + \hat{C}_{i,j}\hat{C}_{t,j} = \sum_{k \neq j}(C_{i,k}C_{t,k}) + \hat{C}_{i,j}C_{t,j} = \begin{cases} [CC^T]_{i,t} - 2 & \text{if } t < j \\ [CC^T]_{i,t} + 2 & \text{if } t > j. \end{cases}$$

We thus get

$$\hat{S}_{i,t} = \begin{cases} S_{i,t} - 1 & \text{if } t < j \\ S_{i,t} + 1 & \text{if } t > j, \end{cases}$$

(remember there is a factor $1/2$ in the definition of $S$). Similarly we get for any $t \neq i$

$$\hat{S}_{j,t} = \begin{cases} S_{j,t} + 1 & \text{if } t < i \\ S_{j,t} - 1 & \text{if } t > i. \end{cases}$$

Finally, for the single corrupted index pair $(i, j)$, we get

$$\hat{S}_{i,j} = \frac{1}{2} \left( n + \sum_{k \neq i,j} \left( \hat{C}_{i,k}\hat{C}_{j,k} \right) + \hat{C}_{i,i}\hat{C}_{j,i} + \hat{C}_{i,j}\hat{C}_{j,j} \right) = S_{i,j} - 1 + 1 = S_{i,j}.$$

For all other coefficients $(s, t)$ such that $s, t \neq i, j$, we have $\hat{S}_{s,t} = S_{s,t}$. Meaning all rows or columns outside of $i, j$ are left unchanged. We first observe that these last equations, together with our assumption that $j - i > 2$ and the fact that the elements of the exact $S$ in (4) differ by at least one, mean that

$$\hat{S}_{s,t} \geq \hat{S}_{s+1,t} \quad \text{and} \quad \hat{S}_{s,t+1} \geq \hat{S}_{s,t}, \quad \text{for any } s < t$$

so $\hat{S}$ remains an R-matrix. Note that this result remains true even when $j - i = 2$, but we need some strict inequalities to show uniqueness of the retrieved order. Indeed, because $j - i > 2$ all these R constraints are strict except between elements of rows $i$ and $i + 1$, and rows $j - 1$ and $j$ (idem for columns). These ties can be broken using the fact that

$$\hat{S}_{i,j-1} = S_{i,j-1} - 1 < S_{i+1,j-1} - 1 = \hat{S}_{i+1,j-1} - 1 < \hat{S}_{i+1,j-1}$$

which means that $\hat{S}$ is still a strict R-matrix (see Figure 1) since $j - 1 > i + 1$ by assumption. ∎

We now extend this result to multiple errors.

**Proposition A.8** *Given all pairwise comparisons $C_{s,t} \in \{-1, 1\}$ between items ranked according to their indices, suppose the signs of $m$ comparisons indexed $(i_1, j_1), \ldots, (i_m, j_m)$ are switched. If the following condition* (10) *holds true,*

$$|s - t| > 2, \text{ for all } s, t \in \{i_1, \ldots, i_m, j_1, \ldots, j_m\} \text{ with } s \neq t, \tag{10}$$

*then $S^{\mathrm{match}}$ defined in* (3) *remains* strict-R, *whereas the score vector $w$ gets $2m$ ties.*

**Proof.** We write the true score and comparison matrix $w$ and $C$, while the observations are written $\hat{w}$ and $\hat{C}$ respectively, and without loss of generality we suppose $i_l < j_l$. This means in particular that $\hat{C}_{i_l,j_l} = -C_{i_l,j_l} = 1$ for all $l$ in $\{1,\ldots,m\}$. To simplify notations we denote by $S$ the similarity matrix $S^{\text{match}}$ (respectively $\hat{S}$ when the similarity is computed from observations).

As in the proof of proposition A.7, corrupted comparisons indexed $(i_l, j_l)$ induce shifts of $\pm 1$ on columns and rows $i_l$ and $j_l$ of the similarity matrix $S^{\text{match}}$, while $S^{\text{match}}_{i_l,j_l}$ values remain the same. Since there are several corrupted comparisons, we also need to check the values of $\hat{S}$ at the intersections of rows and columns with indices of corrupted comparisons. Formally, for any $(i,j) \in \{(i_1,j_1),\ldots(i_m,j_m)\}$ and $t \notin \{i_1,\ldots,i_m,j_1,\ldots,j_m\}$

$$\hat{S}_{i,t} = \begin{cases} S_{i,t}+1 & \text{if } t < j \\ S_{i,t}-1 & \text{if } t > j, \end{cases}$$

Similarly for any $t \notin \{i_1,\ldots,i_m,j_1,\ldots,j_m\}$

$$\hat{S}_{j,t} = \begin{cases} S_{j,t}-1 & \text{if } t < i \\ S_{j,t}+1 & \text{if } t > i. \end{cases}$$

Let $(s,s')$ and $(t,t') \in \{(i_1,j_1),\ldots(i_m,j_m)\}$, we have

$$\hat{S}_{s,t} = \tfrac{1}{2}\left(n + \sum_{k\neq s',t'}\left(\hat{C}_{s,k}\hat{C}_{t,k}\right) + \hat{C}_{s,s'}\hat{C}_{t,s'} + \hat{C}_{s,t'}\hat{C}_{t,t'}\right)$$
$$= \tfrac{1}{2}\left(n + \sum_{k\neq s',t'}\left(C_{s,k}C_{t,k}\right) - C_{s,s'}C_{t,s'} - C_{s,t'}C_{t,t'}\right)$$

Without loss of generality we suppose $s < t$, and since $s < s'$ and $t < t'$, we get

$$\hat{S}_{s,t} = \begin{cases} S_{s,t} & \text{if } t > s' \\ S_{s,t}+2 & \text{if } t < s'. \end{cases}$$

Similar results apply for other intersections of rows and columns with indices of corrupted comparisons (*i.e.* shifts of $0$, $+2$, or $-2$). For all other coefficients $(s,t)$ such that $s,t \notin \{i_1,\ldots,i_m,j_1,\ldots,j_m\}$, we have $\hat{S}_{s,t} = S_{s,t}$. We first observe that these last equations, together with our assumption that $j_l - i_l > 2$, mean that

$$\hat{S}_{s,t} \geq \hat{S}_{s+1,t} \quad \text{and} \quad \hat{S}_{s,t+1} \geq \hat{S}_{s,t}, \quad \text{for any } s < t$$

so $\hat{S}$ remains an R-matrix. Moreover, since $j_l - i_l > 2$ all these R constraints are strict except between elements of rows $i_l$ and $i_l + 1$, and rows $j_l - 1$ and $j_l$ (similar for columns). These ties can be broken using the fact that for $k = j_l - 1$

$$\hat{S}_{i_l,k} = S_{i_l,k} - 1 < S_{i_l+1,k} - 1 = \hat{S}_{i_l+1,k} - 1 < \hat{S}_{i_l+1,k}$$

which means that $\hat{S}$ is still a strict R-matrix since $k = j_l - 1 > i_l + 1$. Moreover, using the same argument as in the proof of proposition A.7, corrupted comparisons induces $2m$ ties in the score vector $w$. ∎

Using similar arguments as above, we study exact ranking recovery conditions with missing comparisons.

**Proposition A.9** *Given pairwise comparisons $C_{s,t} \in \{-1,0,1\}$ between items ranked according to their indices, suppose only one comparison $C_{i,j}$ is missing, with $j - i > 1$ (i.e. $C_{i,j} = 0$), then $S^{\text{match}}$ defined in (3) remains strict-R and the score vector remains strictly monotonic.*

**Proof.** We use the same proof technique as in proposition A.7. We write the true score and comparison matrix $w$ and $C$, while the observations are written $\hat{w}$ and $\hat{C}$ respectively. This means in particular that $\hat{C}_{i,j} = 0$. To simplify notations we denote by $S$ the similarity matrix $S^{\text{match}}$ (respectively $\hat{S}$ when the similarity is computed from observations). We first study the impact of the missing comparison $C_{i,j}$ for $i < j$ on the score vector $\hat{w}$. We have

$$\hat{w}_i = \sum_{k=1}^{n} \hat{C}_{k,i} = \sum_{k=1}^{n} C_{k,i} + \hat{C}_{j,i} - C_{j,i} = w_i + 1,$$

similarly $\hat{w}_j = w_j - 1$, whereas for $k \neq i, j$, $\hat{w}_k = w_k$. Hence, $w$ is still strictly increasing if $j > i + 1$. If $j = i + 1$ there is a tie between $w_i$ and $w_{i+1}$. Now we show that the similarity matrix defined in (3) is a R-matrix. Writing $\hat{S}$ in terms of $S$, we get

$$[\hat{C}\hat{C}^T]_{i,t} = \sum_{k \neq j} \left( \hat{C}_{i,k}\hat{C}_{t,k} \right) + \hat{C}_{i,j}\hat{C}_{t,j} = \sum_{k \neq j} (C_{i,k}C_{t,k}) = \begin{cases} [CC^T]_{i,t} - 1 & \text{if } t < j \\ [CC^T]_{i,t} + 1 & \text{if } t > j. \end{cases}$$

We thus get

$$\hat{S}_{i,t} = \begin{cases} S_{i,t} - \frac{1}{2} & \text{if } t < j \\ S_{i,t} + \frac{1}{2} & \text{if } t > j, \end{cases}$$

(remember there is a factor $1/2$ in the definition of $S$). Similarly we get for any $t \neq i$

$$\hat{S}_{j,t} = \begin{cases} S_{j,t} + \frac{1}{2} & \text{if } t < i \\ S_{j,t} - \frac{1}{2} & \text{if } t > i. \end{cases}$$

Finally, for the single corrupted index pair $(i, j)$, we get

$$\hat{S}_{i,j} = \frac{1}{2} \left( n + \sum_{k \neq i,j} \left( \hat{C}_{i,k}\hat{C}_{j,k} \right) + \hat{C}_{i,i}\hat{C}_{j,i} + \hat{C}_{i,j}\hat{C}_{j,j} \right) = S_{i,j} - 0 + 0 = S_{i,j}.$$

For all other coefficients $(s, t)$ such that $s, t \neq i, j$, we have $\hat{S}_{s,t} = S_{s,t}$. Meaning all rows or columns outside of $i, j$ are left unchanged. We first observe that these last equations, together with our assumption that $j - i > 2$, mean that

$$\hat{S}_{s,t} \geq \hat{S}_{s+1,t} \quad \text{and} \quad \hat{S}_{s,t+1} \geq \hat{S}_{s,t}, \quad \text{for any } s < t$$

so $\hat{S}$ remains an R-matrix. To show uniqueness of the retrieved order, we need $j - i > 1$. Indeed, when $j - i > 1$ all these R constraints are strict, which means that $\hat{S}$ is still a strict R-matrix, hence the desired result. ∎

We can again extend this result to the case where multiple comparisons are missing.

**Proposition A.10** *Given pairwise comparisons $C_{s,t} \in \{-1, 0, 1\}$ between items ranked according to their indices, suppose $m$ comparisons indexed $(i_1, j_1), \ldots, (i_m, j_m)$ are missing, i.e. $C_{i_l, j_j} = 0$ for $i = l, \ldots, m$. If the following condition (11) holds true,*

$$|s - t| > 1 \text{ for all } s \neq t \in \{i_1, \ldots, i_m, j_1, \ldots, j_m\} \tag{11}$$

*then $S^{\text{match}}$ defined in (3) remains strict-R and the score vector remains strictly monotonic.*

**Proof.** Proceed similarly as in the proof of proposition 4.1, except that shifts are divided by two. ∎

We also get the following corollary.

**Corollary A.11** *Given pairwise comparisons $C_{s,t} \in \{-1, 0, 1\}$ between items ranked according to their indices, suppose $m$ comparisons indexed $(i_1, j_1), \ldots, (i_m, j_m)$ are either corrupted or missing. If condition (7) holds true then $S^{\text{match}}$ defined in (3) remains strict-R.*

**Proof.** Proceed similarly as the proof of proposition 4.1, except that shifts are divided by two for missing comparisons. ∎

We now study how frequent are the configurations that allow exact ranking recovery using spectral ranking, in other words how many comparisons can be corrupted before the ranking stops being exact with probability close to one.

**Proposition A.12** *Given a comparison matrix for a set of $n$ items with $m$ corrupted comparisons selected uniformly at random from the set of all possible item pairs. Algorithm SerialRank guarantees that the probability of recovery $p(n, m)$ satisfies $p(n, m) \geq 1 - \delta$, provided that $m = O(\sqrt{\delta n})$. In particular, this implies that $p(n, m) = 1 - o(1)$ provided that $m = o(\sqrt{n})$.*

**Proof.** Let $\mathcal{P}$ be the set of all distinct pairs of items from the set $\{1, 2, \ldots, n\}$. Let $\mathcal{X}$ be the set of all admissible sets of pairs of items, i.e. containing each $X \subseteq \mathcal{P}$ such that $X$ satisfies condition (7). We consider the case of $m \geq 1$ distinct pairs of items sampled from the set $\mathcal{P}$ uniformly at random without replacement. Let $X_i$ denote the set of sampled pairs given that $i$ pairs are sampled. We are interested in the following quantity:

$$p(n, m) = P(X_m \in \mathcal{X}).$$

Given a set of pairs $X \in \mathcal{X}$, let $T(X)$ be the set of nonadmissible pairs, i.e. containing $(i, j) \in \mathcal{P} \backslash X$ such that $X \cup (i, j) \notin \mathcal{X}$.

We have

$$P(X_m \in \mathcal{X}) = \sum_{x \in \mathcal{X}:|x|=m-1} \left(1 - \frac{|T(x)|}{|\mathcal{P}| - (m-1)}\right) P(X_{m-1} = x). \tag{12}$$

Note that every selected pair from $\mathcal{P}$ contributes at most $6n - 10$ nonadmissible pairs, hence, for every $x \in \mathcal{X}$ we have

$$|T(x)| \leq 2(3n-5)|x|.$$

Combined with (12) and the fact $|\mathcal{P}| = \binom{n}{2}$, we have

$$P(X_m \in \mathcal{X}) \geq \left(1 - \frac{2(3n-5)}{\binom{n}{2} - (m-1)}(m-1)\right) P(X_{m-1} \in \mathcal{X}).$$

From this it follows

$$p(n, m) \geq \prod_{i=1}^{m-1} \left(1 - \frac{2(3n-5)}{\binom{n}{2} - (i-1)}i\right)$$

which further implies

$$p(n, m) \geq \prod_{i=1}^{m-1} \left(1 - \frac{i}{a(n, m)}\right)$$

where

$$a(n, m) = \frac{\binom{n}{2} - (m-1)}{2(3n-5)}.$$

Notice that for $m = o(n)$ we have

$$\prod_{i=1}^{m-1} \left(1 - \frac{i}{a(n, m)}\right) \sim \exp\left(-6\frac{m^2}{n}\right) \text{ for large } n.$$

Hence, given $\delta > 0$, $p(n, m) \geq 1 - \delta$ provided that $m = O(\sqrt{n\delta})$. If $\delta = o(1)$, the condition is $m = o(\sqrt{n})$. ∎

## A.4 Numerical Experiments

### A.4.1 England Premier League

We provide results on seasons 2011-2012, 2012-2013 and 2013-2014 in figure 4.

Figure 4: Percentage of upsets (i.e. disagreeing comparisons, lower is better) defined in (8), for various values of $k$ and ranking methods, on England Premier League 2011-2012 season (*left*), 2012-2013 season (*center*), and 2013-2014 season (*right*).