[Reviews · NeurIPS 2014]

Submitted by Assigned_Reviewer_5

This paper proposes an application of seriation algorithm for rank aggregation. They propose a similarity matrix based on pair-wise rankings. They consider two different scenarios for pair-wise rankings and show that seriation algorithm will recover the true ranking (by showing that the proposed similarity matrix is a strict R-matrix). They further provide results on robustness of their method to noise and errors in pairs. They conclude with experiments comparing the new method to BTL and RC algorithm, along with a semi supervised method.

The paper has interesting results both theoretically and application-wise.
Experiments have comparisons and performance illustration; however I have couple of questions:
1- What is the statistical significance of the comparisons given that SD=.02 and the performance differences in figure 2 do not seem to be larger than that in general?
2- In the bottom right plot of figure 2, why is the performance of SR non-monotonically decreasing with m? If this is a simulation error then the plots are not going to be very informative.
3- It is very vague how the semi-supervised ranking is being done, and whether it is part of the contribution of this paper. What is the significance of the semi-supervised method in this paper since it doesn’t connect with the theory and the rest of the paper?
4- Authors seems to consider the official rankings, in the soccer dataset, to be an ideal outcome of their algorithm or any other algorithm, if I am understanding it correctly from last paragraph of the paper. Why are the official rankings the best prediction that an algorithm can have?

The paper is very well written and clear.

The connection of pair-wise similarity and strict R-matrix for the two cases is new as far as I know. Even though the seriation algorithm is a known algorithm, this work is interesting and has an original idea.
The robustness analysis is well done and adds to the technical contribution.

Some suggestions:

1- Proposition 2.4 and its byproducts are limit arguments, where we need large number of observations as an assumption. Looking into their proof they only need the empirical probabilities to keep the same order with the true probabilities. My question is what if P_{ij}s are close, then you will hardly get the same order as the true probabilities in the empirical data. It will be good to talk about how large the sample size should be to have a high certainty on this given minimum of |P_{ij}-P_{ik}| for all i and all j and k (i j k different).

2- The class of models that referred to as GLM, is potentially confusing with the GLM used in linear regression. Even though I see the connection but I have not seen GLM used for this model before. It is probably good to provide a citation which used GLM for this model or use a different name.

3- GLMs in this paper correspond (I think even equivalent) to a class of models called homogeneous Random Utility Models (see, Soufiani et al., Parametric Ranking Models via Rank-Breaking, ICML 14). The connection is interesting because the authors can perform their seriation method on full or partial rankings as well if they use the rank breaking technique proposed in (Soufiani et al, Generalized Method-of-Moments for Rank Aggregation, NIPS13).

Providing a better understanding of the similarity matrices given the R-matrix property and its implications in using seriation algorithms for rank aggregation is an interesting addition to the preference learning literature.

Summary: Overall this is an interesting paper. It is well written and clear. It will be great to add a bit more discussion and clarification in the Numerical Experiments section.

Submitted by Assigned_Reviewer_14

Thanks for the replies to my questions. Of course, I do agree that my example is somewhat specific, although I still believe that it's meaningful with regard the general question regarding the information (not) represented by the similarity relation. I more or less agree on the other answers, although my doubts are not completely dispelled. In summary, I still think the paper puts forward an interesting idea and therefore encourage the authors to continue in this direction, even if the paper might not yet be ready for a publication on the level of the NIPS conference.

Original review:

The idea of reducing the problem of ranking to a problem of similarity analysis appears to be interesting, however, it is not sufficiently well justified in the paper. What exactly is the connection between the preference relation C and the similarity relation S? Do they represent the same information? If not, what information is added or removed, and why is this an advantage?

In the case of a single pairwise preference a > b, represented by the relation

C=
+1 +1
-1 +1 ,

one obtains a symmetric similarity relation, which means that the information about the direction of the preference is completely lost.

Another example: If we have the chain of preferences a > b, b > c, c > d, i.e., the matrix

c= 1 1 0 0
-1 1 1 0
0 -1 1 1
0 0 -1 1 ,

then (unless I made a mistake somewhere) the second-smallest eigenvalue of the Laplacian of S is not unique, and none of the eigenvectors corresponds to the ranking a > b > c > d one would expect to find. The reason could be grounded in the problem that the conversion of preferences into similarities does not capture implicit connections via transitivity. Instead, a non-observed preference contributes to the similarity between two items (by adding 1/2 according to (2)), just like a draw (which, in contrast to a non-observation, is indeed a confirmation of similarity). In this example, the similarity relation is

s =
3.0000 2.0000 1.5000 2.0000
2.0000 3.5000 2.0000 1.5000
1.5000 2.0000 3.5000 2.0000
2.0000 1.5000 2.0000 3.0000

As can be seen, a and d are quite similar, and in fact, a is more similar to d than to c (although it should be the other way around). This is because there is no comparison at all between these items, neither a direct one nor an indirect one via a third item.

If we add the preference d > a to the above chain, we create a cycle, which makes it impossible to produce a reasonable ranking. Again, the second-smallest eigenvalue is not unique, but nevertheless, both Fiedler vectors suggest some ranking. Dies that make sense?

The formal results in section 4 about robustness are technically interesting, however, they refer to very specific cases and compare to very specific alternative methods. For example, it's true that for a single pairwise inversion, the simple Borda count will produce ties. However, there are many other methods that can easily resolve the problem.

Condition (7) is difficult to understand.

The experiments are again interesting, however, I fail to see a clear advantage of the proposed method (for example, in comparison to BTL).

By the way, in connection with the regularity assumptions specified in Definition 2.1 (R-matrix), you might be interested in recent work on ranking based on pairwise preferences induced by statistical rank models such as Mallows or Plackett-Luce. As has been shown, these models imply strong regularity of the pairwise preferences, and the C-matrix they induce is Toeplitz [1,2].

References:

[1] W. Cheng, E. Hüllermeier, W. Waegeman and V. Welker. Label Ranking with Partial Abstention based on Thresholded Probabilistic Models. NIPS-2012, Lake Tahoe, Nevada, USA.

[2] R. Busa-Fekete, E. Hüllermeier, B. Szörenyi. Preference-based rank elicitation: The case of Mallows. ICML 2014.
Summary: In summary, although the idea put forward in the paper is interesting and seems to hold promise, the paper does not yet convince of the reasonableness of the approach (let alone its superiority compared to existing ones).

Submitted by Assigned_Reviewer_39

The paper investigates spectral ranking with pairwise comparisons. The ranking score is given by the Fiedler vector associated with a graph whose similarity weights are given by pairwise comparison data, suggested by Atkins et al., 1998. It can be genearalized to semi-supervised setting using convex relaxation technique in Fogel et al., 2013. Some theoretical properties are established with such a spectral method: 1) in noiseless setting the algorithm recovers the true ranking; 2) in noisy or missing cases, the algorithm exhibits certain stability.

The paper has its spectral ranking algorithm based on Atkins et al., 1998 with some new analysis. It is mathematically sound and clearly written.
Summary: The paper has its spectral ranking algorithm based on Atkins et al., 1998 with some new analysis on exact recovery in noiseless case and robustness in some noisy case.
Author Feedback
Author rebuttal: We would first like to thank the referees for their very constructive comments. We hope to clarify below some of the issues raised in the reports.

Reviewer 14:

- "What exactly is the connection between the preference relation C and the similarity relation S? Do they represent the same information? If not, what information is added or removed, and why is this an advantage?"

The answer to this question depends on the amount of information available. In the round robin tournament setting with all preferences both transitive and available, no information is added nor removed since the underlying total order can be recovered exactly from S using SerialRank. When some comparisons are missing, this one to one correspondence is lost, but the similarity matrix will typically encode more information than scoring methods.

- (About the example for which SerialRank fails): “The reason could be grounded in the problem that the conversion of preferences into similarities does not capture implicit connections via transitivity. Instead, a non-observed preference contributes to the similarity between two items (by adding 1/2 according to (2)), just like a draw (which, in contrast to a non-observation, is indeed a confirmation of similarity).”

This example corresponds to a very specific situation where only a few pairs have a common reference item, which is the exception more than the rule if many pairwise comparisons are given. As judiciously noticed, the similarity matrix defined in (2) has no prior on non-observed preferences, though this could as well be seen as an advantage.

Note that this basic similarity was introduced because of its intuitive and simple form.
While we agree that the similarity matrix defined in (2) is not optimal to handle draws, the paper does propose alternatives which do not have this shortcoming (e.g. in (6) for the GLM), and for which our spectral formulation still applies.

- "The formal results in section 4 about robustness are technically interesting, however, they refer to very specific cases and compare to very specific alternative methods."

We tested against a set of classical references methods. The full catalog of ranking algorithms is too vast to be covered in a conference paper, but these experiments will clearly be expanded in the journal version. Overall, spectral methods have proven invaluable in certain applications (e.g. clustering) and we believe they could prove useful for ranking too. Probabilistic methods work very well but algorithmic diversity is always helpful, this is our key motivation here.

- We thank the reviewer for the very interesting references provided, they were added to the paper.

Reviewer 5:

- “What is the statistical significance of the comparisons given that SD=.02 and the performance differences in figure 2 do not seem to be larger than that in general?”

Given that SD=.02 and that the experiments were repeated 50 times, the 95% confidence interval of the expected Kendall tau has a half-width of 1.96*0.02/sqrt(50)=0.0055, which means that on average, SerialRank performs better.

- “In the bottom right plot of figure 2, why is the performance of SR non-monotonically decreasing with m? If this is a simulation error then the plots are not going to be very informative.”

There is no simulation error. If local uncertain comparisons (i.e. |i-j| < m) were equal to 0, we would recover the true ranking up to m=50. The fact that local comparisons are random noise, combined with robustness properties of the spectral algorithm, seems to induce the complex behavior around m=50.

- “It is very vague how the semi-supervised ranking is being done, and whether it is part of the contribution of this paper. What is the significance of the semi-supervised method in this paper since it doesn’t connect with the theory and the rest of the paper?”

We mention semi-supervised ranking because it is a clear benefit of the seriation formulation. We cannot discuss it at length here, but the paper shows how to turn semi-supervised ranking problems into semi-supervised seriation problems, which are discussed in details in [Fogel et al., 2013].

- “Authors seem to consider the official rankings, in the soccer dataset, to be an ideal outcome of their algorithm or any other algorithm, if I am understanding it correctly from last paragraph of the paper. Why are the official rankings the best prediction that an algorithm can have?”

Official rankings are not the best rankings in terms of loss, but we expect a good ranking not to be too far from the official one, hence we use them as a natural benchmark. All unsupervised results are naturally subjective.

- “Proposition 2.4 and its byproducts are limit arguments, where we need large number of observations as an assumption. Looking into their proof they only need the empirical probabilities to keep the same order with the true probabilities. My question is what if P_{ij}s are close, then you will hardly get the same order as the true probabilities in the empirical data. It will be good to talk about how large the sample size should be to have a high certainty on this given minimum of |P_{ij}-P_{ik}| for all i and all j and k (i j k different).”

This indeed needs to be quantified and we are working on a more formal argument about robustness for the journal version.

- “The class of models that referred to as GLM, is potentially confusing with the GLM used in linear regression. Even though I see the connection but I have not seen GLM used for this model before. It is probably good to provide a citation which used GLM for this model or use a different name.

Yes indeed. A reference to [Mc Cullagh, P. and Nelder, Generalized Linear Models, 1989] was added to the paper. We also thank the reviewer for his suggestion of connecting our work with the rank breaking technique in [Soufiani et al, Generalized Method-of-Moments for Rank Aggregation, NIPS13].